# Risk Stratification for ECMO Requirement in COVID-19 ICU Patients Using Quantitative Imaging Features in CT Scans on Admission

**DOI:** 10.3390/diagnostics11061029

**Published:** 2021-06-03

**Authors:** Eva Gresser, Jakob Reich, Bastian O. Sabel, Wolfgang G. Kunz, Matthias P. Fabritius, Johannes Rübenthaler, Michael Ingrisch, Dietmar Wassilowsky, Michael Irlbeck, Jens Ricke, Daniel Puhr-Westerheide

**Affiliations:** 1Department of Radiology, University Hospital, LMU Munich, 81377 Munich, Germany; jakob.reich@med.uni-muenchen.de (J.R.); bastian.sabel@med.uni-muenchen.de (B.O.S.); wolfgang.kunz@med.uni-muenchen.de (W.G.K.); matthias.fabritius@med.uni-muenchen.de (M.P.F.); johannes.ruebenthaler@med.uni-muenchen.de (J.R.); michael.ingrisch@med.uni-muenchen.de (M.I.); jens.ricke@med.uni-muenchen.de (J.R.); daniel.puhr-westerheide@med.uni-muenchen.de (D.P.-W.); 2Department of Anesthesiology, University Hospital, LMU Munich, 81377 Munich, Germany; dietmar.wassilowsky@med.uni-muenchen.de (D.W.); michael.irlbeck@med.uni-muenchen.de (M.I.)

**Keywords:** COVID-19, respiratory distress syndrome, extracorporeal membrane oxygenation, artificial intelligence, computed tomography scan

## Abstract

(1) Background: Extracorporeal membrane oxygenation (ECMO) therapy in intensive care units (ICUs) remains the last treatment option for Coronavirus disease 2019 (COVID-19) patients with severely affected lungs but is highly resource demanding. Early risk stratification for the need of ECMO therapy upon admission to the hospital using artificial intelligence (AI)-based computed tomography (CT) assessment and clinical scores is beneficial for patient assessment and resource management; (2) Methods: Retrospective single-center study with 95 confirmed COVID-19 patients admitted to the participating ICUs. Patients requiring ECMO therapy (*n* = 14) during ICU stay versus patients without ECMO treatment (*n* = 81) were evaluated for discriminative clinical prediction parameters and AI-based CT imaging features and their diagnostic potential to predict ECMO therapy. Reported patient data include clinical scores, AI-based CT findings and patient outcomes; (3) Results: Patients subsequently allocated to ECMO therapy had significantly higher sequential organ failure (SOFA) scores (*p* < 0.001) and significantly lower oxygenation indices on admission (*p* = 0.009) than patients with standard ICU therapy. The median time from hospital admission to ECMO placement was 1.4 days (IQR 0.2–4.0). The percentage of lung involvement on AI-based CT assessment on admission to the hospital was significantly higher in ECMO patients (*p* < 0.001). In binary logistic regression analyses for ECMO prediction including age, sex, body mass index (BMI), SOFA score on admission, lactate on admission and percentage of lung involvement on admission CTs, only SOFA score (OR 1.32, 95% CI 1.08–1.62) and lung involvement (OR 1.06, 95% CI 1.01–1.11) were significantly associated with subsequent ECMO allocation. Receiver operating characteristic (ROC) curves showed an area under the curve (AUC) of 0.83 (95% CI 0.73–0.94) for lung involvement on admission CT and 0.82 (95% CI 0.72–0.91) for SOFA scores on ICU admission. A combined parameter of SOFA on ICU admission and lung involvement on admission CT yielded an AUC of 0.91 (0.84–0.97) with a sensitivity of 0.93 and a specificity of 0.84 for ECMO prediction; (4) Conclusions: AI-based assessment of lung involvement on CT scans on admission to the hospital and SOFA scoring, especially if combined, can be used as risk stratification tools for subsequent requirement for ECMO therapy in patients with severe COVID-19 disease to improve resource management in ICU settings.

## 1. Introduction

Since its onset in December 2019, the severe acute respiratory syndrome coronavirus 2 (SARS-CoV-2) pandemic has become a global challenge for healthcare systems, particularly due to limited resources of intensive care units (ICU). In 2020, Coronavirus disease 2019 (COVID-19) disease climbed to the third leading cause of death in the US according to the Centers for Disease Control [1]. Around 15–30% of COVID-19 inpatients require intensive care treatment, 15–20% require intubation and a substantial subpopulation of around three quarters of ICU patients develop respiratory failure such as acute respiratory distress syndrome (ARDS) [2,3,4,5,6,7,8]. In severe hypoxemic respiratory failure, extracorporeal membrane oxygenation (ECMO) can be a valuable lifesaving bridging technique providing time for potential organ recovery or, in rare cases, lung transplant [9,10,11,12]. Veno-venous ECMO (VV-ECMO) is indicated in severe hypoxemic respiratory failure refractory to conventional respiratory support such as low pressure and low tidal volume mechanical ventilation with optimal positive end expiratory pressure (PEEP), neuromuscular blockade and prone positioning [13,14,15]. Patients exhibiting cardiac or circulatory failure might be assigned to veno-arterial ECMO (VA-ECMO) for additional circulatory support independent of the extent of respiratory failure.

In severe cases of COVID-19 with refractory hypoxemia the use of ECMO as a rescue therapy has been advocated [16,17,18]. Published ECMO mortality rates have ranged widely from around 40 to as high as 90% or above, whereas increasing evidence suggests that COVID-19 ECMO mortality might be similar to known ARDS ECMO mortality rates of around 40–60% and might not be significantly different to overall COVID-19 ICU mortality [2,8,12,17,18,19,20,21,22,23]. Because ECMO therapy might reduce mortality and outcome is likely to improve when therapy is applied early in severe ARDS, early risk stratification and patient allocation is crucial [17,24,25,26]. This might also be applicable for patients with severe COVID-19 pneumonia. In this study we evaluated the potential of clinical parameters on ICU admission as well as AI-based CT imaging features on hospital admission for risk stratification of ECMO therapy in critically ill COVID-19 patients.

## 2. Materials and Methods

### 2.1. Patient Data

Our retrospective single-center study was approved by the local institutional review board. All COVID-19 patients (*n* = 95) admitted from 03/2020 until 01/2021 and already discharged or deceased by end of January 2021 with positive SARS-CoV-2 PCR testing and computed tomography (CT) scans within 48 h of hospital admission to the two participating ICUs, which have been dedicated to exclusive COVID-19 care during the pandemic, were included in the study (Figure 1). Patient data were collected retrospectively and extracted from our digital patient information system (QCare PDMS, Health Information Management GmbH, Bad Homburg, Germany), which is routinely used at the corresponding ICUs: e.g., age, gender, body mass index (BMI), length of stay on ICU, hospital discharge or death, hours of invasive and non-invasive ventilation, re-intubation, sequential organ failure (SOFA) score, respiratory data as oxygenation indices, lung compliances and PEEP values. Chest X-ray (CXR) and chest CTs were extracted from the digital radiologic information system (RIS) and picture archiving and communication system (PACS).

### 2.2. Image Acquisition

CT scans (*n* = 91) were performed using CT scanners of our emergency department (Siemens Somatom Force, Somatom AS+ and GE Optima 660), either as non-contrast high-resolution scan or with contrast-enhanced pulmonary embolism protocol with the patient in supine position. Image acquisition was modulated between 80 and 120 kVp with adaptive tube current (mAS). All images were reconstructed with slice thicknesses of 1.00 mm or 1.25 mm. Multiplanar reconstruction methods were performed on all images. CT scans for *n* = 4 patients were performed at external hospitals before transfer to our hospital for ICU therapy with comparable scanning parameters. The datasets were suitable for AI-assessment and included in the present study.

### 2.3. Artificial Intelligence Based Quantification of Lung Involvement

The CAD4COVID CT report tool (Thirona B.V., Nijmegen, The Netherlands) was used for the quantification of CT lung involvement under the supervision of two radiologists with 4 and 7 years of clinical experience, respectively. CAD4COVID provides segmentation of lung lobes and displays them through a colored heatmap. The affected lung volume is quantified as percentage of the total lung volume (0–100%) and a score is generated ranging between 0 and 25 which indicates the extent of COVID-19 related abnormalities on the CT scan (0–5 points per lobe, maximum score 25 overall, Figure 2). The performance of the CAD4COVID method in the detection of COVID-19 was rated comparable with that of human readers, as shown in an evaluation study [27]. CAD4COVID is a freely usable CE-certified tool (class II, CE 0344) and access can be requested via the Thirona website (URL Thirona website). It is made available free-of-charge to support healthcare facilities during the pandemic. Axial lung kernel CT scans can be uploaded in DICOM file format after anonymization.

### 2.4. Prediction Parameters for the Regression Analysis

The demographic characteristics age, sex and body mass index (BMI) have been shown to be significant risk factors for disease severity and were therefore included in our regression model [6,28]. As ECMO represents a rescue therapy for patients with severely disturbed blood oxygenation capabilities due to lung damage, the oxygenation index on admission was also selected as an important admission parameter for evaluation. Lactate on admission as a general parameter for shock and the SOFA score on admission as a multiparametric indicator of organ failure were included in the analyses. SOFA score rates six different organ systems on a scale of zero to four points (range 0 to 24 points). Additionally, the overall affected area as a percentage of the total lung volume of the CAD4COVID tool was used as imaging features for the prediction model.

### 2.5. Statistical Analysis

All statistical analyses were performed with SPSS software (version 26.0, IBM). Continuous variables are reported as median with interquartile ranges (IQR). Mann–Whitney-U for continuous variables and Chi-square test or Fisher’s exact test for categorical variables were applied to test for differences between the standard ICU therapy and the ECMO therapy groups. Significance was defined as a two-sided *p*-value < 0.05. Binary logistic regression for the prediction of allocation to ECMO therapy was performed adjusting for multiple covariates. Odds ratios with 95% confidence intervals are shown. Receiver operating characteristic (ROC) analyses using exact binomial confidence intervals (CI) were used to compare the predictive performance of parameters and the area under the curve (AUC) was calculated. Ideal discriminative values were determined using maximization of the Youden index and sensitivity as well as specificity are reported.

## 3. Results

### 3.1. Baseline Clinical Characteristics and Demographic Data

Of the 95 patients, 78% included in the study were male, median age was 66 years (IQR 55–74), median BMI was 27 (IQR 25–33), median SOFA score on admission was 8 (IQR 5–11), median lactate on admission was 1.3 (IQR 1.0–1.8) and median oxygenation index on admission was 168 (IQR 112–229). Patients were classified according to the Berlin definition for ARDS, whereas 2 patients (2.4%) did not exhibit ARDS on admission, 24 (28.9%) presented with mild, 40 (48.2%) with moderate and 17 (20.5%) with severe ARDS features on admission. Median CT severity score was 15 (IQR 10–20) and median percentage of lung involvement was 36% (IQR 19–56). All baseline characteristics are shown in Table 1.

### 3.2. Differences between the ECMO Group and ICU Standard Therapy Group

Fourteen of the 95 COVID-19 patients (14.7%) required ECMO therapy, 12 patients were allocated to VV-ECMO (86%) and 2 patients to VA-ECMO (14%). Patients treated with ECMO had a median age of 62 years (IQR 55–68) vs. 68 years (IQR 55–75) in the standard ICU therapy group, *p* = 0.164. Sex was equally distributed between the groups with 79.0% male in the ECMO group versus 71.4% male in the standard ICU group, *p* = 0.528. BMI was significantly higher in the ECMO group with a median of 31 (IQR 27–37) vs. 27 (IQR 25–30), *p* = 0.031. Patients in the ECMO group were significantly more often diagnosed with severe ARDS according to the Berlin definition, *p* = 0.029.

Median length of ICU stays for patients receiving ECMO therapy was 22.3 (IQR 8.4–29.1) vs. 12.5 (IQR 5.5–23.9) days on ICU, *p* = 0.120. For the survivors of both groups, median length of stay was significantly longer for ECMO patients with 51.9 (IQR 39.6–64.2) vs. 12.1 (IQR 5.5–20.6) days on ICU, *p* = 0.014. In the standard ICU therapy group, 73% of patients received mechanical ventilation, in the ECMO-therapy group 100% of patients were mechanically ventilated. Hours on the ventilator were significantly longer for patients in the ECMO group compared to the standard care group (median hours on the ventilator 516.0 (IQR 192.9–698.5) vs. 157.6 (IQR 0.0–401.1), *p* = 0.003) as well as only including survivors from both groups with 1012.4 (IQR 946.7–1078.1) vs. 113.9 (IQR 0.0–327.2), *p* = 0.014). Hours of non-invasive ventilation (NIV) were significantly shorter for ECMO patients compared to the control group (0.8 (IQR 0.0–25.3) vs. 3.3 (IQR 0.0–12.3), *p* = 0.006). Significantly more patients were treated with hemodiafiltration in the ECMO group (13 patients, 92.9%) vs. the standard ICU group (25 patients, 30.9%; *p* < 0.001), median hours on hemodiafiltration were 143.5 (IQR 41.7–346.5) in the ECMO group vs. 90.5 (IQR 34.0–280.2), *p* = 0.361. In the ECMO group 8 patients (57.1%) were treated with prone positioning for lung recruitment with a median of 22.5 h (IQR 18.0–56.4) vs. 25 patients (30.9%) with a median of 23.5 h (IQR 15.5–36.0) in the standard ICU therapy group, no statistically significant differences were detected. In the ECMO group, SOFA score on admission, mean SOFA score during stay and maximum SOFA score during stay were significantly higher than in the standard ICU therapy group (12 (IQR 10–14) vs. 8 (IQR 4–11), *p* < 0.001, Figure 3A; 14.5 (IQR 12.5–18.8) vs. 7.5 (IQR 5.1–10.6), *p* < 0.001 and 18 (IQR 15–22) vs. 12 (IQR 8–15), *p* < 0.001, respectively). Further, the oxygenation index was significantly lower in the ECMO group on admission (110 (IQR 90–161) vs. 178 (IQR 121–232), *p* = 0.009). Imaging on hospital admission showed a significantly higher severity score (21 (IQR 19–22) vs. 14 (IQR 10–19), *p* < 0.001) and significantly higher lung volume involvement (66% (IQR 49–72) vs. 30% (IQR 17–53), *p* < 0.001, Figure 3B) in the AI based CT assessment.

Patients in the ECMO therapy group exhibited a significantly longer time interval from admission to the time point of maximum SOFA score (13 days (IQR 2–15) vs. 2 days (IQR 1–8) in the standard ICU therapy group, *p* = 0.012). This might be explained by a longer disease progression reaching significantly higher SOFA scores during the course of the disease for patients in the ECMO therapy group. The increase of SOFA score per day until reaching the maximum SOFA score did not differ significantly between groups (*p* = 0.836). Time from admission to death did not differ significantly between groups for non-survivors (*p* = 0.932). Median time from admission to ECMO allocation and placement was 1.4 days (IQR 0.2–4.0). The in-hospital mortality differed significantly between groups with 85.7% non-survivors in the ECMO therapy group vs. 29.6% non-survivors in the standard ICU therapy groups, mirroring the significant clinical differences on admission and confirming the use of ECMO therapy as a last resort for patients with the most severe COVID-19 disease progression. In multivariate binary logistic regression for mortality, the ECMO group was not significantly associated with a higher mortality after adjustment for clinical and demographic parameters (Appendix A). All results of the comparison between groups with corresponding *p*-values can be obtained from Table 2.

### 3.3. Risk Stratification for ECMO Therapy

In multivariate binary logistic regression analysis for the prediction of allocation to ECMO therapy during treatment on ICU including the parameters age, sex, BMI, SOFA score, lactate, oxygenation index and AI based assessment of CT imaging on hospital admission, only SOFA score and CT imaging findings on hospital admission were significantly associated with ECMO allocation during the subsequent treatment in ICU (odds ratio for SOFA score 1.32 (95% CI 1.08–1.62), *p* = 0.008 and for lung involvement on CT 1.06 (95% CI 1.01–1.11), *p* = 0.011), results are shown in Table 3. Additionally, we performed a multivariate binary logistic regression analysis for the prediction of allocation to ECMO therapy excluding the two patients with VA-ECMO yielding similar results (Appendix A). We also evaluated the predictive potential of comorbidities for ECMO therapy allocation but did not find a significant association (Appendix A). Using receiver operating characteristic (ROC) curves, we found an area under the curve (AUC) of 0.83 (95% CI 0.73–0.94) for lung involvement in percent of total lung volume on CT imaging on admission and an AUC of 0.82 (95% CI 0.72–0.91) for SOFA score on ICU admission (Figure 4). A combined parameter (multiplication of SOFA score on admission with percentage of lung involvement on admission CT) yielded the best predictive results with an AUC of 0.91 (95% CI 0.84–0.97, Figure 4). For a combination score of 435 (best discriminative value) we calculated a Youden index of 0.77 with a sensitivity of 93% and a specificity of 84% (Table 4).

## 4. Discussion

Early identification of ECMO therapy requirements for COVID-19 patients with insufficient oxygenation capacity might reduce mortality and improve outcome after hospitalization, especially when applied timely during disease progression [17,24,25,26]. Accessible and reliable risk stratification as early as possible is crucial for further ICU therapy planning and effective resource management to assure optimal treatment for severely affected COVID-19 patients. However, as ECMO is an immensely resource-intensive approach and requires scarce capacities in specialized maximum care centers at high expenses, adequate triage of patients is of utmost importance and must meet high requirements [29,30].

We analyzed clinical data and quantitative CT imaging features of 95 SARS-CoV-2 PCR-positive ICU patients at our hospital. AI-based quantification of lung involvement as percentage of the total lung volume in COVID-19 ICU patients on admission CT could predict ECMO requirement with an AUC of 0.83 (CI 0.73–0.94). Further, the SOFA score on admission as a parameter for organ failure showed a substantial predictive power yielding an AUC of 0.82 (95% CI 0.72–0.91). As ECMO is a bridging technique for patients with severe oxygenation impairment and severe COVID-19 often manifests with ARDS it is comprehensible that lung involvement is a decisive factor. When the SOFA score on admission, as a measure for multi organ function, was taken into account and severity of lung involvement was weighted with the SOFA score in a combined prediction model, predictive capability could even be improved. The combined parameter (multiplication of SOFA on ICU admission and lung involvement on admission CT) showed the best discriminative potential with an AUC of 0.91 (95% CI 0.84–0.97) and a sensitivity of 0.93 and a specificity of 0.84 was calculated with a Youden index of 0.77. While lung involvement alone showed a high specificity whereas SOFA scoring alone displayed a high sensitivity, specificity could substantially be increased by quantitative CT imaging features in the combined model while a high sensitivity was preserved. Other parameters included in the analyses that have previously been associated with increased risk for a severe course of the disease such as age, gender and body mass index did not show discriminative power for allocation to ECMO therapy [6,28].

The role of CT in diagnosis, triage and allocation purposes for COVID-19 patients was acknowledged early in the pandemic [31]. CT scans as a noninvasive and widely available tool have been shown to be of value in risk stratification in COVID-19 [32,33,34,35]. However, reading of CT scans is often done manually by radiologists which is time consuming, especially if segmentation and quantitative evaluation needs to be done, and is subjective with an inter- as well as intra-observer variability. Artificial intelligence (AI) is therefore increasingly important for supporting radiologic workups and has been shown to promisingly derive quantitative CT imaging features. In previous studies AI has been shown to accurately predict lung cancer [36,37] and outcomes of ARDS [38]. In COVID-19 pneumonia, deep learning was used to differentiate COVID-19 disease from community-acquired pneumonia [39] and CT quantification of pneumonia lesions in COVID-19 CT features was used to predict severe disease course based on changes in chest CT scans from day 0 to day 4 [40]. However, risk stratification for ECMO therapy based on chest CT scans on hospital admission in combination with clinical features has not yet been reported and seems promising, as AI-derived features from CTs at an early stage of COVID-19 can be used to predict progression to severe oxygenation impairment with the requirement of a potentially lifesaving bridging therapy. A recent multi-center study with a larger VV-ECMO COVID-19 patient cohort (exclusively investigating VV-ECMO patients) found that the SOFA score was not predictive for survival of patients when collected right before initiation of the VV-ECMO and thus at a very critical stage of the disease course with severe oxygenation impairment [41]. In our study, the SOFA score on admission was not predictive for in-hospital mortality in the overall group. However, our results indicate that the SOFA score on admission to ICU is valuable for the assessment of the COVID-19 patient’s risk for ECMO therapy.

### Limitations

First, this study is a retrospective analysis with a limited sample size due to a finite number of COVID-19 patients with severe disease that could be treated in our hospital on the participating ICUs. Furthermore, discharged patients were not followed up beyond their hospital stay. Nevertheless, our hospital represents one of the largest maximum care university hospitals in Europe and 95 patients with severe COVID-19 disease were included in this single center analysis. The study results need further investigation, ideally on larger external validation cohorts. Second, the overall in-hospital mortality and particularly for patients allocated to ECMO therapy was very high in this study. This might be explained by a cohort with unusually high disease severity (high clinical scores for disease severity on admission). In order to transfer results from our study to other ICU settings and hospitals, an external validation cohort including less severe COVID-19 ICU patients is desirable.

## 5. Conclusions

AI-based quantitative assessment of lung volume involvement on admission CT, particularly if combined with the sequential organ failure assessment score, is a non-invasive and easily accessible tool to support risk stratification of ECMO requirements in severely ill COVID-19 patients upon ICU admission and can assist in early patient assessment and resource management.

## Figures and Tables

**Figure 1 diagnostics-11-01029-f001:**
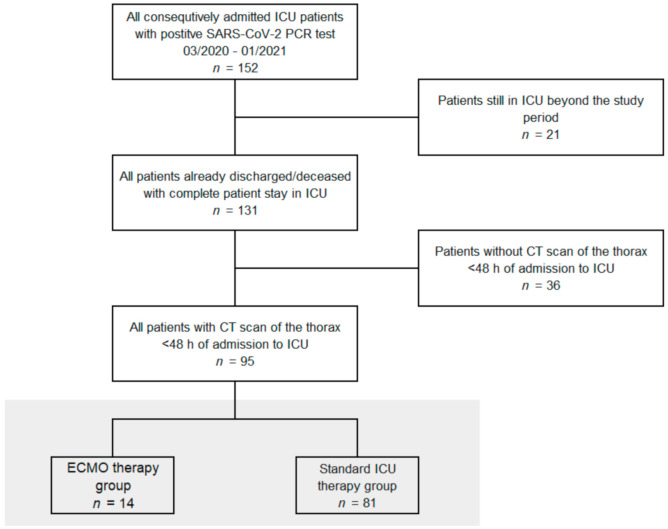
Study flow chart. All patients from the participating intensive care units (ICUs) with positive severe acute respiratory syndrome coronavirus 2 (SARS-CoV-2) polymerase chain reaction (PCR) test between 2nd of March 2020 and 26th of January 2021, who were discharged or deceased and received a computed tomography (CT) scan of the thorax on admission, were included. Extracorporeal membrane oxygenation, ECMO.

**Figure 2 diagnostics-11-01029-f002:**
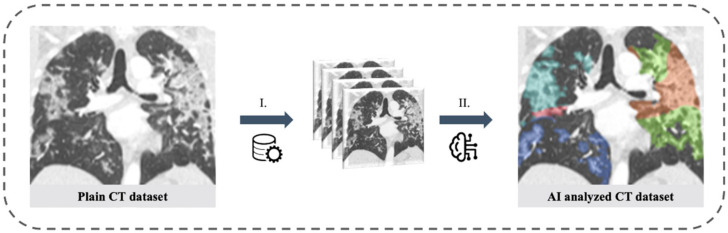
Lung involvement of CT scans of the thorax assessed with artificial intelligence. Left: Plain coronal slice image of the thorax CT scan in a patient with SARS-CoV-2 positive PCR test. Right: AI analyzed CT dataset with color staining of affected lung tissue (colored areas in each lung lobe). I. Pre-processing with data anonymization and uploading. II. AI-based analysis of CT dataset.

**Figure 3 diagnostics-11-01029-f003:**
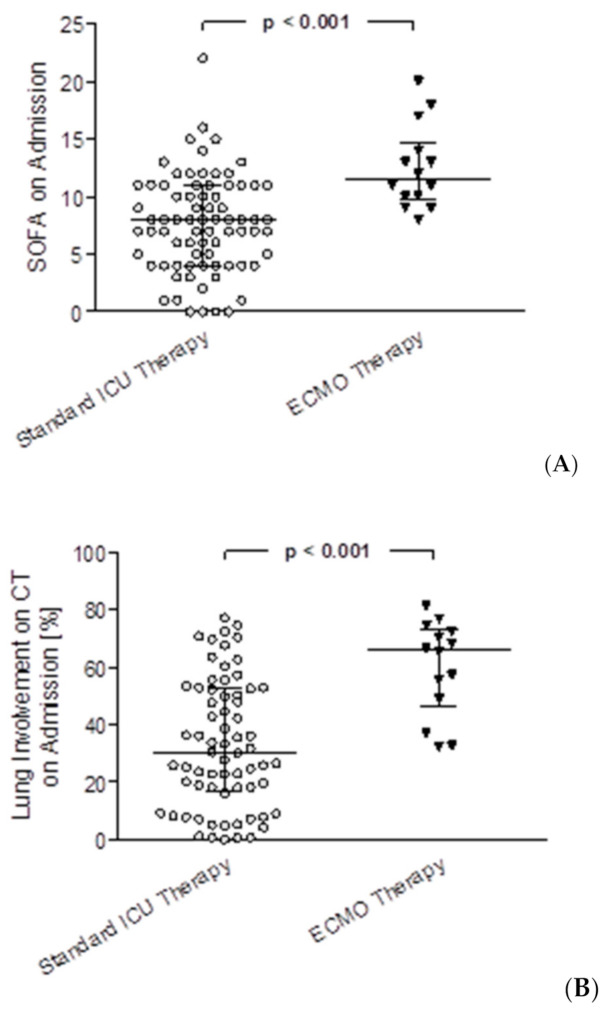
SOFA score on admission and lung involvement on admission CT scans assessed with artificial intelligence for COVID-19 patients with standard ICU therapy vs. ECMO therapy. (**A**) Comparison of SOFA scores on admission between patients with standard ICU therapy vs. ECMO therapy, *p* < 0.001 (**B**) comparison of lung involvement on admission CT scans between patients with standard ICU therapy vs. ECMO therapy, *p* < 0.001.

**Figure 4 diagnostics-11-01029-f004:**
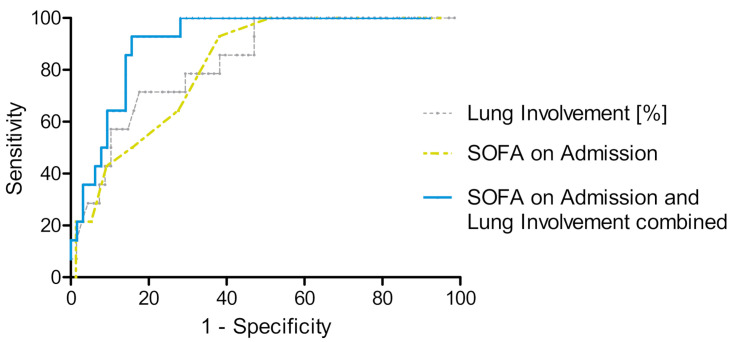
ROC curves for SOFA score on ICU admission, lung involvement on admission CT and both parameters combined for the prediction of subsequent ECMO therapy. ROC curve for assignment to ECMO therapy according to SOFA score on admission (yellow), AUC = 0.82 (95% CI 0.72–0.91), lung involvement in % of total lung volume (grey), AUC = 0.83 (95% CI 0.73–0.93) and SOFA score on ICU admission combined with lung involvement on admission CT (blue), AUC = 0.91 (95% CI 0.84–0.97).

**Table 1 diagnostics-11-01029-t001:** Characteristics of COVID-19 ICU patients. Baseline characteristics for included COVID-19 patients admitted to the participating ICUs. Values presented are count (percentage) for categorical and median (interquartile range) or mean (± standard deviation) for ordinal or continuous variables. ICU, intensive care unit; ARDS, acute respiratory distress syndrome; COVID-19, Coronavirus disease 2019; SOFA score, sequential organ failure assessment score; CT, computed tomography; * 5 missing values; ** 13 missing values.

	COVID-19 ICU-patients (*n* = 95)	
**Patient Data**		
Age	66	(55–74)
Male Sex	74	(77.9%)
Body Mass Index	27	(25–33)
SOFA Score on Admission *	8	(5–11)
Lactate on Admission	1.3	(1.0–1.8)
Oxygenation Index on Admission **	168	(110–229)
**Comorbidities**		
Diabetes	33	(34.7%)
Hypertension	59	(62.1%)
Heart Disease	32	(33.7%)
Pulmonary Disease	19	(20.0%)
Chronic Kidney Disease	9	(9.5%)
Active Malignancy	10	(10.5%)
Immunosuppression	7	(7.4%)
**ARDS Type on Admission ****		
Mild	25	(30.1%)
Moderate	40	(48.2%)
Severe	15	(18.1%)
No ARDS on Admission	3	(3.6%)
**CT Features on Admission ****		
CT-Severity Score	15	(10–20)
CT-Percentage of Lung Involvement	36	(19–56)

**Table 2 diagnostics-11-01029-t002:** COVID-19 ARDS patients with standard care vs. ECMO-therapy. Comparison of baseline parameters, clinical parameters during ICU stay and outcome parameters between the standard ICU therapy group and the ECMO therapy group. Values presented are count (percentage) for categorical and median (interquartile range) or mean (± standard deviation) for ordinal or continuous variables. ICU, intensive care unit; ARDS, acute respiratory distress syndrome; COVID-19, Coronavirus disease 2019; SOFA score, sepsis-related organ failure assessment score; CT, computed tomography; ECMO, extracorporeal membrane oxygenation; * 6 missing values in standard ICU therapy group; ** 5 missing values in standard ICU therapy group; *** 13 missing values in standard ICU therapy group.

	Standard ICU Therapy (*n* = 81)		ECMO Therapy (*n* = 14)		*p* Value
**Comparison of Patient Characteristics**					
Age	68	(55–75)	62	(55–68)	*p* = 0.164
Male Sex	64	(79.0%)	10	(71.4%)	*p* = 0.528
BMI	27	(25–30)	31	(27–37)	*p* = 0.031
**ARDS Type**					
Mild	7	(9.3%)	0	(0.0%)	n/a
Moderate	34	(45.3%)	2	(14.3%)	*p* = 0.006
Severe	34	(45.3%)	12	(85.7%)	*p* = 0.029
**Patient Data during ICU Stay**					
Days on ICU (including external ICUs)	12.5	(5.5–23.9)	22.3	(8.4–29.1)	*p* = 0.120
Days on ICU (Survivors)	12.1	(5.5–20.6)	51.9	(39.6–64.2)	*p* = 0.014
Number of Patients on Mechanical Ventilation	59	(72.8%)	14	(100%)	*p* = 0.026
Hours on Ventilator	157.6	(0.0–401.1)	516.0	(192.9–698.5)	*p* = 0.003
Hours on Ventilator (Survivors)	113.9	(0.0–327.2)	1012.4	(946.7–1078.1)	*p* = 0.014
Hours on NIV	3.3	(0.0–12.3)	0.8	(0.0–25.3)	*p* = 0.006
Number of Patients with HDF	25	(30.9%)	13	(92.9%)	*p* < 0.001
Hours on Hemodiafiltration	90.5	(34.0–280.2)	143.5	(41.7–346.5)	*p* = 0.361
Prone Position	25	(30.9%)	8	(57.1%)	*p* = 0.057
Hours of Prone Position	23.5	(15.5–36.0)	22.5	(18–56.4)	*p* = 0.636
SOFA mean*	7.5	(5.1–10.6)	14.5	(12.5–18.8)	*p* < 0.001
SOFA max**	12	(8–15)	18	(15–22)	*p* < 0.001
SOFA on Admission*	8	(4–11)	12	(10–14)	*p* < 0.001
Oxygenation Index on Admission***	178	(121–232)	110	(90–161)	*p* = 0.009
CT Severity Score on Admission***	14	(10–19)	21	(19–22)	*p* < 0.001
CT Percentage of Lung Involvement on Admission***	30	(17–53)	66	(49–72)	*p* < 0.001
**Disease Progression**					
Time from Admission to SOFA max	2	(1–8)	13	(2–5)	*p* = 0.012
Time from Admission to Death	17	(5–28)	19	(7–23)	*p* = 0.932
Time from Admission to ECMO Placement (days)	n/a		1.4	(0.2–4.0)	n/a
Delta SOFA from admission to max per day	0.8	(0.0–4.0)	0.5	(0.3–2.0)	*p* = 0.836
Time from CT to ICU Admission (days)	1	(1–3)	1	(0–1)	*p* = 0.078
Time from CT to ECMO Placement (days)	n/a		2.5	(1–5)	n/a
Time from Hospital Admission to ICU Admission	1	(1–4)	1	(1–1)	*p* = 0.119
**Outcome**					
In-hospital mortality	24	(29.6%)	12	(85.7%)	*p* < 0.001

**Table 3 diagnostics-11-01029-t003:** Predictors for ECMO therapy for COVID-19 ICU-patients. Results from binary logistic regression with adjustment for multiple covariates. ECMO, extracorporeal membrane oxygenation; COVID-19, coronavirus disease 2019; ICU, intensive care unit; CI, confidence interval; BMI, body mass index; SOFA, sequential organ failure assessment; CT, computed tomography; 18 patients not included in regression analysis due to missing values. * *p* < 0.05.

	ECMO Therapy		
Independent Variables	Odds Ratio	CI	*p* Value
Age	1.003	0.924–1.089	0.936
Sex	0.509	0.064–4.073	0.525
BMI	1.065	0.934–1.214	0.350
SOFA on Admission	1.320	1.077–1.617	0.008 *
Lactate on Admission	0.991	0.528–1.859	0.977
CT Lung Involvement (%) on Admission	1.059	1.013–1.106	0.011 *

**Table 4 diagnostics-11-01029-t004:** ROC analysis for the prediction of ECMO therapy with Youden index, sensitivity and specificity. ICU, intensive care unit; ECMO, extracorporeal membrane oxygenation; AUC, area under the curve; Y-index; Youden index; SOFA, sequential organ failure assessment score; CT, computed tomography. *n* = 81 Standard ICU therapy, *n* = 14 ECMO therapy.

Standard ICU Therapy (*n* = 68) vs. ECMO Therapy (*n* = 14)	AUC (95% CI)		Y-Index	Discriminative Value	Sensitivity	Specificity
SOFA Score on Admission	0.82	0.72–0.91	0.50	8.5	0.93	0.57
Standard ICU Therapy (*n* = 76) vs. ECMO Therapy (*n* = 14)						
Lung Involvement on CT (%)	0.83	0.73–0.93	0.54	55.7	0.71	0.82
Standard ICU Therapy (*n* = 64) vs. ECMO Therapy (*n* = 14)						
SOFA Score on Admission and Lung Involvement on CT (%) combined	0.91	0.84–0.97	0.77	435	0.93	0.84

## Data Availability

The datasets analyzed during the current study are available from the corresponding author on reasonable request. The used CAD4COVID tool to analyze the CT data sets is a CE-certified tool which is made available freely by Thirona B.V., Nijmegen, Netherlands (URL https://thirona.eu/cad4covid/, accessed on 3 June 2021).

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
