# Peer review of "Risk Stratification for ECMO Requirement in COVID-19 ICU Patients Using Quantitative Imaging Features in CT Scans on Admission"

_diagnostics, 2021, doi:10.3390/diagnostics11061029_

Round 1
Reviewer 1 Report
A multicentric retrospective study was recently published (Supady A et al., Outcome Prediction in Patients with Severe COVID-19 Requiring Extracorporeal Membrane Oxygenation-A Retrospective International Multicenter StudyMembranes (Basel). 2021 Feb 27;11(3):170. doi: 10.3390/membranes11030170),. In this study, where a larger number of patients were included, the authors did not observed any significant propective value for SOFA score. This should be discussed
Reviewer 2 Report
The authors describe the use of AI-analyzed CT scans from COVID-19 patients in risk stratification for need for ECMO therapy. The manuscript is clear and well written. The findings are novel and interesting to the readers of Diagnostics. I however have the following suggestions for improvement.
Major comments
- There is a significant overlap in AI-CT and in SOFA scores (and I presume in the combined score too) between ECMO and non-ECMO patients, which makes it difficult to separate the groups based on these variables. Also, indications for ECMO have been defined in the ELSO guidelines, so while AI-based CT assessment can certainly have a valuable role in risk stratification, it is not correct to say that it is "crucial for patient management" when it comes to prediction of/indication for ECMO. I suggest the authors acknowledge this in their discussion, and consider rephrasing line 15 ("is crucial for patient management") and line 35 (" allocation management") in the abstract . I do think the increased specificity with preserved sensitivity, after addition of AI-CT score to SOFA score, deserves to be mentioned in the discussion.
- Time from CT scan to ICU admission and from CT scan to ECMO initiation is not reported, and is relevant to judge the prognostic value of CT. I would include the time from admission to ECMO initiation in the abstract.
- I would do a sensitivity analysis excluding the 2 VA ECMO patients (unless they also had severe respiratory failure), as chest CT severity is less likely to be predictive of need for circulatory mechanical support. Also, as oxygenation index calculation requires p/F ratio, and SOFA respiratory score is based on p/F ratio, it might be better to use an adapted SOFA score without the respiratory subscore in the model where both SOFA and oxygenation index are included.
Minor comments
- abstract, line 12: severely affected lungs, rather than damaged (which suggest all COVID-19-related changes are permanent)
- introduction, line 42: SARS-CoV2 signifies severe acute respiratory syndrome coronavirus 2
- introduction, line 45-46: CDC abbreviation with capital C, D and C
- introduction, ine 63-64: outcome is more likely to improve when therapy is applied early. Add a reference for this statement, as data for ECMO outcome based on duration of mechanical ventilation in COVID-19 patients have been published.
- Introduction, line 67: is the CT within 48h of hospital admission, or within 48h of ICU admission? It would be relevant (also in Results) to report the time between hospital admission and ICU admission.
- Materials and Methods, image acquisition, line 98. I presume that for ventilated patients, the CT scan was not taken at end-inspiration, as this is often hard to time.
- Materials and Methods, line 102: transfer (not transferral)
- Materials and Methods, subheading 2.4. Why were age, sex, BMI, oxygenation index, lactate, SOFA included in the model, but not (1) comorbidities such as hypertension, diabetes, chronic kidney disease – or the Charlson score, and not (2) other variables that have been associated with severe disease course e.g. lymphopenia, D-dimers and myocardial injury? Also, the authors should consider adding pCO2 to the variables included in the prognostic model.
- Results: it would be relevant to know how many patients were not on invasive mechanical ventilation.
